# New Insights into the Antimicrobial Action of Cinnamaldehyde towards *Escherichia coli* and Its Effects on Intestinal Colonization of Mice

**DOI:** 10.3390/biom11020302

**Published:** 2021-02-18

**Authors:** Wellison A. Pereira, Carlos Drielson S. Pereira, Raíssa G. Assunção, Iandeyara Savanna C. da Silva, Fabrícia S. Rego, Leylane S. R. Alves, Juliana S. Santos, Francisco Jonathas R. Nogueira, Adrielle Zagmignan, Thomas T. Thomsen, Anders Løbner-Olesen, Karen A. Krogfelt, Luís Cláudio N. da Silva, Afonso G. Abreu

**Affiliations:** 1Laboratório de Patogenicidade Microbiana, Programa de Pós-Graduação em Biologia Microbiana, Universidade Ceuma, São Luís 65075-120, Brazil; well.ap@usp.br (W.A.P.); drielsonn.sousa@gmail.com (C.D.S.P.); raissa_guara@hotmail.com (R.G.A.); savannacarsi@gmail.com (I.S.C.d.S.); fabricia_sr@hotmail.com.br (F.S.R.); leylanesusy@hotmail.com (L.S.R.A.); julianass98@hotmail.com (J.S.S.); frjonathas@outlook.com (F.J.R.N.); adrielle.zagmignan@ceuma.br (A.Z.); luiscn.silva@ceuma.br (L.C.N.d.S.); 2Programa de Pós-Graduação em Ciências da Saúde, Universidade Federal do Maranhão, São Luís 65080-805, Brazil; 3Department of Functional Genomics, University of Copenhagen, 2200 Copenhagen, Denmark; thomas.thomsen@bio.ku.dk (T.T.T.); lobner@bio.ku.dk (A.L.-O.); 4Department of Science and Environment, Roskilde University, 4000 Roskilde, Denmark; kak@ssi.dk

**Keywords:** cinnamaldehyde, intestinal colonization, natural products

## Abstract

*Escherichia coli* is responsible for cases of diarrhea around the world, and some studies have shown the benefits of cinnamaldehyde in the treatment of bacterial disease. Therefore, the objective of this study was to evaluate the effects of cinnamaldehyde in mice colonized by pathogenic *E. coli*, as well as to provide more insights into its antimicrobial action mechanism. After determination of minimum inhibitory (MIC) and minimum bactericidal (MBC) concentrations, the interference of cinnamaldehyde in macromolecular pathways (synthesis of DNA, RNA, protein, and cell wall) was measured by incorporation of radioisotopes. The anti-adhesive properties of cinnamaldehyde towards *E. coli* 042 were evaluated using human epithelial type 2 (HEp-2) cells. Intestinal colonization was tested on mice, and the effect of cinnamaldehyde on *Tenebrio molitor* larvae. Cinnamaldehyde showed MIC and MBC values of 780 μg/mL and 1560 μg/mL, respectively; reduced the adhesion of *E. coli* 042 on HEp-2 cells; and affected all the synthetic pathways evaluated, suggesting that compost impairs the membrane/cell wall structure leading bacteria to total collapse. No effect on the expression of genes related to the SOS pathway (*sulA* and *dinB1*) was observed. The compound did not interfere with cell viability and was not toxic against *T. molitor* larvae. In addition, cinnamaldehyde-treated mice exhibited lower levels of colonization by *E. coli* 042 than the untreated group. Therefore, the results show that cinnamaldehyde is effective in treating the pathogenic *E. coli* strain 042 and confirm it as a promising lead molecule for the development of antimicrobial agents.

## 1. Introduction

*Escherichia coli* is an important pathogen responsible for numerous cases of diarrhea worldwide, representing a serious problem for immunocompromised individuals, and especially children [1,2,3,4]. Several reports have associated diarrhea with significant delays in childhood development [1,3,5].

In a study carried out in South America, Africa and Asia, in children and adults with diarrhea, the predominant pathogen isolated in fecal samples was enteroaggregative *E. coli* (EAEC), a pathotype of diarrheagenic *E. coli* [6,7]. Depending on the region, EAEC can be the etiologic agent of up to 30% of episodes of diarrhea in infants and young children, as well as in adults with persistent diarrhea [8].

According to the World Health Organization, antibacterial drugs have become less effective or even ineffective, resulting in an accelerating global health security emergency that is rapidly outpacing available treatment options [9]. Therefore, due to the difficult treatment of several diseases of microbial origin, it is important to identify and characterize compounds of natural origin that can be used safely in the treatment of infections. Cinnamaldehyde has been used as a potential alternative for antimicrobial therapy by several in vitro and in vivo studies [10,11]. It is a major component found in the essential oil extracted from cinnamon bark (*Cinnamomum cassia*; *Lauraceae*), being responsible for the characteristic taste and odor of the species [10,12].

Cinnamaldehyde has shown a wide spectrum of antimicrobial activity by inhibiting pathogens such as *Candida* spp. [13,14], *E. coli* [15], *Listeria monocytogenes* [16], *Pseudomonas aeruginosa* [17], and *Staphylococcus aureus* [18]. Most of the studies carried out with this compound are related to the food industry, where it is especially used for its antimicrobial properties [12]. Recently, the oral supplementation with cinnamaldehyde was able to inhibit the colonization of uropathogenic *E. coli* (UPEC) in lower urinary tract infection in mice [11]. Besides, it is pointed as a promising agent to treat inflammatory disorders [19,20], diabetes [21], and cancer [22,23]. Therefore, the aim of this study was to evaluate the effect of cinnamaldehyde in colonization of mouse gut by pathogenic *E. coli*, as well as to provide more insights into its antimicrobial action mechanism.

## 2. Material and Methods

### 2.1. Bacteria

*E. coli* strains were used to evaluate the effect of cinnamaldehyde in this study (Table 1). The strains were kept at −80 °C in trypticase soy broth (TSB) plus 20% glycerol and grown in Luria–Bertani broth (LB), LB agar, or MacConkey, plus appropriate antibiotics when indicated.

The reporter strain *E. coli* ALO 4696 was obtained by P1 transduction using phage lysates of the *E. coli* ALO4025 (MG1655; *sfiA::Km*) into *E. coli* ALO3980 (MG1655; *sfiA*::lacZ). The transductants were selected using agar plates containing kanamycin (10 µg/mL). The expression of *sulA* can be quantified in the resultant strain (ALO 4696) as it is integrated with β-galactosidase gene (reporter gene) and the deletion of *seqA* prevents excessive filamentation. The expression of *sulA* was tested using various concentrations of ciprofloxacin after different incubation time. β-galactosidase activities were measured as described by Miller [30].

### 2.2. Minimum Inhibitory (MIC) and Minimum Bactericidal Concentration (MBC)

Overnight cultures of *E. coli* strains were diluted (1:100) in Mueller Hinton broth (MH) and grown until they reached an optical density of 0.1 at 600 nm. Aliquots of 10 μL of this suspension were added to wells containing different concentrations of cinnamaldehyde (ranging from 195 μg/mL to 6240 μg/mL). Cinnamaldehyde (*trans*-cinnamaldehyde 99%) used in this study was obtained from Sigma-Aldrich (Darmstadt, Germany).

After 24 h incubation at 37 °C, resazurin 0.03% was used to determine the minimum inhibitory concentration (MIC). Aliquots from those wells without bacterial growth were transferred to MacConkey agar plates to determine the minimum bactericidal concentration (MBC). All assays were performed in triplicate with at least two repetitions [31].

### 2.3. Macromolecular Synthesis 

The effects of cinnamaldehyde on the assembly of key bacterial cellular processes was evaluated by testing cinnamaldehyde’s effect on macromolecular synthesis: incorporation of radioactive precursors [methyl-^3^H] thymidine, uridine, arginine and glucosamine for synthesis of DNA, RNA, protein, and cell wall, respectively (Table 2).

For this, *E. coli* MG1655 was exponentially grown in minimum medium supplemented with 2.5 mg thiamine/mL and 0.5% (*w*/*v*) glucose (ABTG) until it reached an OD_450_ of 0.1. At OD_450_ of 0.1, growing cultures were split into several flasks depending on treatment. The cultures were treated with cinnamaldehyde (1000 µg/mL) or antibiotic (Table 2) after 35 min. Samples (500 μL) and OD_450_ were collected at determined periods (15, 35, 50, 60, 80, 100 and 120 min). Macromolecular incorporation was measured by addition of 0.375 μCi of each precursor to the 500 µL sample. The sample was allowed to grow at 37 °C, according to the specific time of incorporation of each precursor (Table 2). After this period, 5 mL of 5% TCA containing 0.1 M NaCl was used to stop the reaction, followed by the reading of OD_450_ of the sample culture. The samples were filtered, followed by filter washing (two times) with 5% TCA. Finally, the filters were packed into scintillation tubes for overnight drying. After this step 5 mL scintillation fluid (ULTIMA GOLD, PerkinElmer, Waltham, MA, USA) was added to the vials and labeled precursors were quantified using a scintillation counter HIDEX 300 SL (Turku, Finland). Thus, the value obtained by reading the radioactivity in counts per minute (CPM) was divided by the OD_450_ to account for differences in growth rate in incorporation rate.

### 2.4. Evaluation of the Expression of Genes Associated with SOS Response

The effect of cinnamaldehyde on SOS response was performed using *E. coli sulA:: lacZ* (ALO 4696) and *E. coli dinB1::lacZ* (ALO 562) (Table 1). Overnight cultures of both strains were diluted in LB broth (1:100) and grown until OD_600_ of 0.1 was reached. Strains were incubated with ciprofloxacin as positive control (½ MIC and ¼ MIC; 0.5 μg/mL and 0.25 μg/mL, respectively) and cinnamaldehyde (½ MIC and ¼ MIC) at 37 °C under shaking at 150 rpm. After 3 h, 500 μL of cell suspension were permeabilized with 100 μL of Toluene (Merck, Darmstadt, Germany). Following, the supernatant (100 μL) was added to 1 mL of the Zeta buffer containing ONPG (substrate for the enzyme β-Galactosidase). The tubes were again incubated in a water bath at 30 °C with shaking, and the time that each sample took until the color change was verified. After color change the reaction was stopped with a solution of sodium bicarbonate followed by reading of OD at 450 nm. 

### 2.5. Adhesion Test with Human Epithelial Type 2 (Hep-2) Cells

The adherence test was carried out according to the protocol described by Scaletsky et al. [32] with modifications. HEp-2 cells were cultured in 50 mL bottles (Nunc, Inter Med, Roskilde, Denmark) containing Dulbeco’s Modified Minimum Eagle Medium (DMEM) containing antibiotics penicillin (100 U/mL) and streptomycin (1 mg/mL) (Cultilab, Campinas, Brazil), plus 10% fetal bovine serum for 2–3 days at 37 °C under an atmosphere of 10% O_2_ and 90% CO_2_. After this incubation period, cells were transferred to Nunc 24-well plates (Merck, Darmstadt, Germany) containing glass coverslips and incubated under the same conditions until reaching 75% confluence.

*E. coli* 042 was cultured in 3 mL of LB without shaking at 37 °C for 18 h, and aliquots of 20 μL of bacterial cultures were added to each well of the plate containing HEp-2 cells. Following 3 h of incubation, cinnamaldehyde (600 μg/mL) was added to each well. After washing and fixation steps with methanol, cells were stained with methylene blue eosin dye in May–Grünwald solution (Merck, Darmstadt, Germany) and azur-eosin-methylene blue dye solution according to Giemsa (Merck). After washing for removal of excess dye, the coverslips were dried at room temperature and mounted on slides for Entellan (Merck) microscopy. Then, the slides were analyzed by light microscopy.

### 2.6. Cytotoxicity Assay

The assay is based on the extent of the damage induced by the compound. It is used to determine cell viability by quantifying the MTT (3-(4,5-dimethylthiazol-2-yl)-2,5-diphenyltetrazolium bromide) present in the medium reduced by the cellular metabolic activity bound to NADH forming blue formazan crystals [33].

To evaluate a possible cytotoxic effect of cinnamaldehyde, the compound at various concentrations (from 100 μg/mL to 12,000 μg/mL) was incubated with Vero or Hep-2 cells for a period of 48 h. After the incubation period, 100 μL of MTT was added to each well and the plates incubated for 3 h at 37 °C and 5% CO_2_. Then, 100 μL of dimethylsulfoxide (DMSO) was added and the samples were homogenized for complete dissolution of the formazan crystals, before measuring absorbance at 550 nm.

### 2.7. Effect of Cinnamaldehyde on Tenebrio molitor Larvae

The objective of this step was to evaluate a possible toxic effect of the compound on *T. molitor* larvae prior to animal testing. Therefore, larvae were randomly selected (~200 mg) for toxicity tests (*n* = 10/group). In the survival test, one group of lavas was inoculated with 10 μL of phosphate buffered saline (PBS) and another with 10 μL of cinnamaldehyde at a concentration corresponding to 10× MIC. After inoculation, the larvae were incubated at 37 °C and the mortality rate was observed for 7 days. Kaplan–Meier curve and the long-rank test were used for the survival analysis [34].

### 2.8. Mouse Colonization

We performed animal studies on female swiss mice obtained from the Central Animal House of the Ceuma University (São Luís, Brazil). The mice were ten to twelve weeks old, average body weight ~25 g and maintained at 26 ± 2 °C, 44% to 56% relative humidity, under 12 h light-dark cycles, and maintained with free access to sterile food and acidified water. Bacterial inoculation was performed by gavage described below. All procedures were assessed and approved by the Committee of Ethics in Research of the Ceuma University (Process nº 229/17).

The animals were separated into four groups with six animals each: PBS; animals infected with *E. coli* 042; animals infected with *E. coli* 042 and treated with cinnamaldehyde 20 mg/kg; and animals infected with *E. coli* 042 and treated with cinnamaldehyde 40 mg/kg.

The streptomycin-treated mouse model [35] was used to investigate the intestinal colonization by EAEC 042, as well as the treatment with 200 μL daily of cinnamaldehyde (20 or 40 mg/kg) by gavage, after the colonization.

Initially, Swiss mice provided ad libitum with drinking water containing 5 g/liter of streptomycin from 48 h prior to the inoculation and for the duration of the experiment were used. Bacterial suspensions of EAEC 042 were prepared at a final concentration of 5 × 10^3^ CFU/mL and 200 µL of these suspensions was administered orogastrically by gavage. Fresh fecal samples were collected in sterile tubes, weighted, diluted, and homogenized in sterile PBS. Serial dilutions of these preparations (1:10^1^ until 1:10^6^) were then plated onto MacConkey agar containing streptomycin (100 µg/mL) for determination of CFU/g. Bacteria were quantified by plate counts for 15 consecutive days post infection. PCR for detection of *pic* (virulence marker) was also performed to confirm intestinal colonization. Primer sequences, amplified product size, and annealing temperature for *pic* is described by Abreu et al. [36].

### 2.9. Statistical Analysis

Statistical analyzes were performed using Graph Pad Prism software, version 7.0. The results were expressed as mean and standard deviation and were subjected to ANOVA, followed by the multiple comparisons test from Tukey’s test, *T*-test or Kruskal–Wallis, and Mann–Whitney tests when the data normality assumption was not satisfied. The Kaplan–Meier curve and the long-rank test (*p* < 0.05) were used for survival analysis.

## 3. Results

### 3.1. Cinnamaldehyde Inhibits E. coli Growth 

The first step of this research was to evaluate the antimicrobial action of cinnamaldehyde against *E. coli* strains. The compound showed MIC ranging from 780 to 3120 µg/mL among the strains (Table 3). For both *E. coli* 042 (reference prototype for intestinal colonization studies) and HB101 (non-pathogenic), the MIC and MBC values were 780 and 1560 µg/mL, respectively. The strain *E. coli* MG1655 was killed by cinnamaldehyde at 390 µg/mL.

We also evaluated the antimicrobial effect of cinnamaldehyde in a strain derived from *E. coli* BW25113 (*E. coli* ALO 4628). However, we did not observe significant changes on MIC values for cinnamaldehyde towards these strains in relation to the wild type (Table 3).

### 3.2. Cinnamaldehyde Interferes in the Macromolecular Synthesis in E. coli

In order to provide more insights into cinnamaldehyde’s effects we assessed the interference of cinnamaldehyde in the synthesis of DNA, RNA, protein and glucosamine (a component of cell wall) in *E. coli*. As observed in Figure 1, after the addition of cinnamaldehyde, all pathways were impaired.

Since cinnamaldehyde blocked DNA synthesis, we evaluated whether cinnamaldehyde treatment is related to the activation of SOS response. For this purpose, two strains of *E. coli* with SOS-related genes fused with *lacZ* (ALO4696 *sulA::lacZ* and ALO562 *dinB1::lacZ*) were used (Table 1). As expected, the expression of both genes was upregulated by ciprofloxacin (positive control) (Figure 2). However, cinnamaldehyde did not induce the expression of these genes, suggesting that this compound did not directly inhibit DNA replication. In this sense, the blockage in DNA synthesis is expected to be a secondary effect of cinnamaldehyde, possibly by disrupting cellular homeostasis by action on the cell membrane.

### 3.3. Cinnamaldehyde Does Not Interfere with Cell Viability and Is Not Toxic against T. molitor Larvae

In order to assess whether cinnamaldehyde would be viable for the in vivo tests, we analyzed its toxic potential towards VERO and Hep-2 cells. After 48 h of incubation, the substance did not induce significant differences on cell viability (Figure 3). Similarly, cinnamaldehyde did not show toxicity towards *T. molitor* larvae (Figure 4).

### 3.4. Cinnamaldehyde Promotes Reduction of EAEC 042 Aggregative Adhesion on HEp-2 Cells

It was verified whether cinnamaldehyde was able to inhibit the adhesion of the EAEC 042 strain onto HEp-2 cells (Figure 5). It was possible to observe that the compound was able to inhibit the adhesion of strain 042 after 3 h at a concentration of 600 µg/mL. As can be seen in Figure 5, when comparing the adhesion pattern of EAEC 042 and EAEC 042 + treatment with cinnamaldehyde, it can be noted the substance’s anti-adhesion effect on the strain tested.

### 3.5. Cinnamaldehyde Treatment Reduced the Intestinal Colonization of Mice by E. coli

The ability of EAEC 042 to colonize the intestine mucosa was evaluated using the streptomycin-treated mouse model. The strain was able to colonize mice on the second day post infection and up to the fifteenth day, with a peak of colonization observed on the eighth day. From the eighth to the fourteenth day, colonization declined, probably in response to immune system action (Figure 6).

Testing two different cinnamaldehyde concentrations, we showed that the group receiving daily doses of cinnamaldehyde 40 mg/kg presented a decrease in the colonization on the fourth day, unlike the untreated group. The group treated with cinnamaldehyde 20 mg/kg also showed a significant reduction after the sixth day when compared to the untreated group. From the eighth to the last day, the decrease in the count of the microorganisms remained with slight variation in both groups (20 and 40 mg/kg) (Figure 6).

## 4. Discussion

In this study, we analyzed the antimicrobial activity of cinnamaldehyde against *E. coli* strains and provided new insights into its antimicrobial action mechanism. We showed that cinnamaldehyde was able to inhibit *E. coli* growth, conforming previous studies. He et al. [15] evaluated the action of cinnamaldehyde on *E. coli*, showing that it has an inhibitory effect on the growth. Wang et al. [10] evaluated cinnamaldehyde on the biofilm formation of *Porphyromonas gingivalis*. The authors report that the compound inhibits formation of biofilm even at sub inhibitory concentrations.

We also assessed cinnamaldehyde toxicity on epithelial cells and in *T. molitor* larvae. The experiment showed that cinnamaldehyde was non-toxic to Hep-2 and VERO cells, nor was it detrimental to *T. molitor* larvae. Similarly, Ferro et al. [18] emphasized not only the antimicrobial power of the compound, but also brought new data about its protective character in a *Galleria mellonella* larvae model, widely used in toxicity tests. In addition, the authors showed that cinnamaldehyde presented a bactericidal action against *Staphylococcus aureus* and multi-resistant *Enterococcus faecalis*, as well as increased larval survival and reduced the amount of *S. aureus* isolated in larval hemolymph.

After the toxicity assays, we evaluated whether cinnamaldehyde treatment could be related to activation of SOS response. Superoxide dismutase (SOD) has been associated with defense against reactive oxygen species and bacterial resistance to damage caused by antimicrobial substances [15]. In this study, *E. coli* strains 4554 (*sodA*) and 4555 (*sodB*) were exposed to stress conditions promoted by exposure to cinnamaldehyde. We could not confirm the results by He et al. [15], which correlates the superoxide dismutase genes with the action of cinnamaldehyde. For the authors, the total SOD action of the *E. coli* strain tested is proportional to the increase in cinnamaldehyde concentration, indicating that the compound may lead to oxidative damage of the cell membrane and also to SOD activity, which would lead to a more effective response and, consequently, resistance, especially in the antioxidant gene SOD.

When we analyzed the SOS response using *sulA* and *dinB1* genes, the data indicated that there was a low expression of these genes, showing that cinnamaldehyde interferes in the bacteria growth and causes death without necessarily inducing an SOS.

The *dinB* gene was first described in 1980 as part of the DNA damage response. It was shown to be induced by exposure to UV radiation [37]. According to Ordonez et al. [38], *dinB* (also called DNA Polymerase IV) has its transcription controlled by *lexA*, when there is damage to the genetic material. It has also been associated with repetitive sequence replication and mutation [39]. Suttom et al. [40], when reviewing the aspects related to SOS response, affirmed that *dinB* would not be associated to repairs for damages caused by UV radiation, but to tolerance to other types of DNA damage.

Furthermore, about 30 genes linked to *E. coli* SOS response are induced after damage to bacterial DNA, among which, *sulA* is one of the most important [40]. The expression of *sulA* is an indication of stress, since it increases after DNA damage, and may serve as a basis for studies aimed at evaluating the mode of action of antimicrobial candidates. Moreover, the *sulA* product acts to prevent cell division when it is not in favorable conditions [41].

However, once it was known that cinnamaldehyde did not induce an SOS response by any of the two routes analyzed, it was necessary to evaluate the site of action of the compound. For this, several tests using radioisotopes were performed to evaluate the macromolecular incorporation of precursors for the assembly of essential structures of the bacteria. It was observed that cinnamaldehyde prevented incorporation of the four precursors evaluated. It is noteworthy that the compound had a faster effect by preventing the incorporation of glucosamine and consequently, no cell wall formation.

The action of cinnamaldehyde on the cell membrane has previously been described in the literature. Wang et al. [10], after analyzing the effect of the substance on the biofilm formed by *P. gingivalis*, investigated the morphological changes, cell membrane damage and DNA, RNA, and protein breakdown. The authors also affirmed that increasing the dose caused greater damage, the morphological changes were irreversible and caused loss of membrane integrity. Furthermore, RNA and DNA synthesis were inhibited. Likewise, He et al. [15] tested the inhibitory effect, oxidative, and membrane damage caused by cinnamaldehyde on *E. coli* ATCC 25922. By use of Raman spectroscopy the authors showed that the compound had a negative impact on the wall by causing loss of cellular constituents.

In addition to cell damage, a possible interference with the pathogen’s ability to adhere is an important assessment that needs to be made. Thus, cell tests are used for a variety of purposes in research laboratories, such as diagnosing, classifying, and identifying *E. coli* adhesion patterns, for example. Among the different pathotypes, EAEC 042 is one of the most studied for its pathogenicity. Most of these studies differentiate EAEC 042 from the other *E. coli* pathotypes by tests with HEp-2 cells. It is known that it has, as its main characteristic, the high adhesion capacity, attributed to its virulence factors, mainly fimbriae and other aggregative proteins [5].

It is important to note that *E. coli* 042 adhesion characteristics are standard for EAEC pathotype identification. Jensen et al. [5] described it as having an aggregative adhesion character, binding both to the cell and to other bacteria. Didactically, this form of adhesion is compared to “stacked bricks” and leads to the formation of biofilm, an important structure for maintaining pathogen reservoirs and infection progress.

Here, we demonstrated that cinnamaldehyde was able to reduce EAEC 042 aggregative adhesion on HEp-2 cells. This finding corroborates what was described by Ferro et al. [18], in which cinnamaldehyde reduced the adhesion of *S. aureus*. Likewise, Prabuseenivasan and colleagues [42], when analyzing the in vitro effect of several essential oils, demonstrated that most of them had antibacterial action against *E. coli*. Similarly, in a recent study by Li et al. [43], the antimicrobial action of cinnamaldehyde was demonstrated. The authors emphasized the potency of the compound primarily against *S. aureus* ATCC25923, *Bacillus subtilis* ATCC 9372, *E. coli* ATCC 25922, and *Pseudomonas aeruginosa* ATCC 27853.

Since cinnamaldehyde inhibited growth of several *E. coli* strains and interfered in the macromolecular synthesis, as well as promoted a reduction of EAEC 042 aggregative adhesion on HEp-2 cells, we investigated whether the compound also had the ability to reduce mouse intestinal colonization by EAEC 042.

Swiss mice were infected with EAEC 042 and after establishment of colonization, groups of animals were treated daily by gavage with the compound at two different concentrations in order to verify the action of cinnamaldehyde. At the end of the 15 days of colonization, both groups treated daily with cinnamaldehyde (20 and 40 mg/kg) were able to reduce colonization drastically as early as the sixth and eighth day, respectively, leading to the number of CFUs close to zero on the last day, in this way, showing a better performance against the strain of *E. coli* tested.

Other research has also demonstrated the antimicrobial action of cinnamaldehyde over *E. coli*. Narayanan et al. [11] in vivo study evaluated oral oil treatment in C57BL mice with urinary tract infection. At the end of the experiment, it was possible to affirm that the treatment reduced colonization of the bacterium in the bladder and urethra and that it was not toxic to the animals.

In a recent study, Yuan and Yuk [44] evaluated the possible interference of some essential oils on the virulence of *E. coli* O157:H7, an important cause of gastrointestinal infection, in non-lethal concentrations. They demonstrated that the use of cinnamaldehyde led to the temporary suppression of motility, reduced biofilm-forming ability and did not provoke the resistance of *E. coli* strain tested. The authors emphasized that this result indicates that the compound can be used for its antimicrobial properties in the food industry.

Malheiro et al. [45] evaluated cinnamaldehyde on three important pathogens, *E. coli* NCTC (MIC 3 mM)*, S. aureus,* and *Enterococcus hirae,* comparing their results with those of biocides; as in the present study, the researchers found that there was a reduction in microbial growth and that cinnamaldehyde was able to inhibit *E. coli* NCTC at low concentrations. In turn, Yuan et al. [44] decided to investigate the possible interference of natural compounds on bacterial growth using the Time-Kill method, thus it was observed that cinnamaldehyde has good antimicrobial action, especially when combined with eugenol, another component of cinnamon essential oil; the damage they caused to the membrane was considered the main cause.

Firmino and colleagues [46] evaluated the antimicrobial power of the compound on biofilm, a microbial structure important for disease establishment and bacterial protection. According to the authors, cinnamaldehyde was able to inhibit biofilm formation, both in Gram-positive and Gram-negative bacteria, such as *E. coli*. On the other hand, Field et al. [47] examined the antimicrobial potential of cinnamaldehyde together with another natural compound, bacteriocin Nisin, in a solution with ethylenediaminetetraacetic acid (EDTA) against enterotoxigenic *E. coli*. At the end of the experiments, it was found that the combination increased the observed antimicrobial potential compared to the results found in the tests with any of the compounds alone.

Taken together our data point to an important role of cinnamaldehyde as inhibitor of the growth of the pathogenic strain EAEC 042 in vitro and in vivo tests. The compound was not toxic to *T. molitor* larvae or Hep-2 and Vero cells, and reduced the adhesion of the bacterium on HEp-2 cells. In addition, it was possible to show that the compound interfered in the incorporation of key molecules for the assembly of essential structures to the bacterium. Finally, intestinal colonization of mice by EAEC 042 was reduced with cinnamaldehyde treatment (20 and 40 mg/kg). Such results show that the compound is effective in the treatment against the pathogenic strain 042 and a promising candidate for the development of novel antibacterial drugs.

## Figures and Tables

**Figure 1 biomolecules-11-00302-f001:**
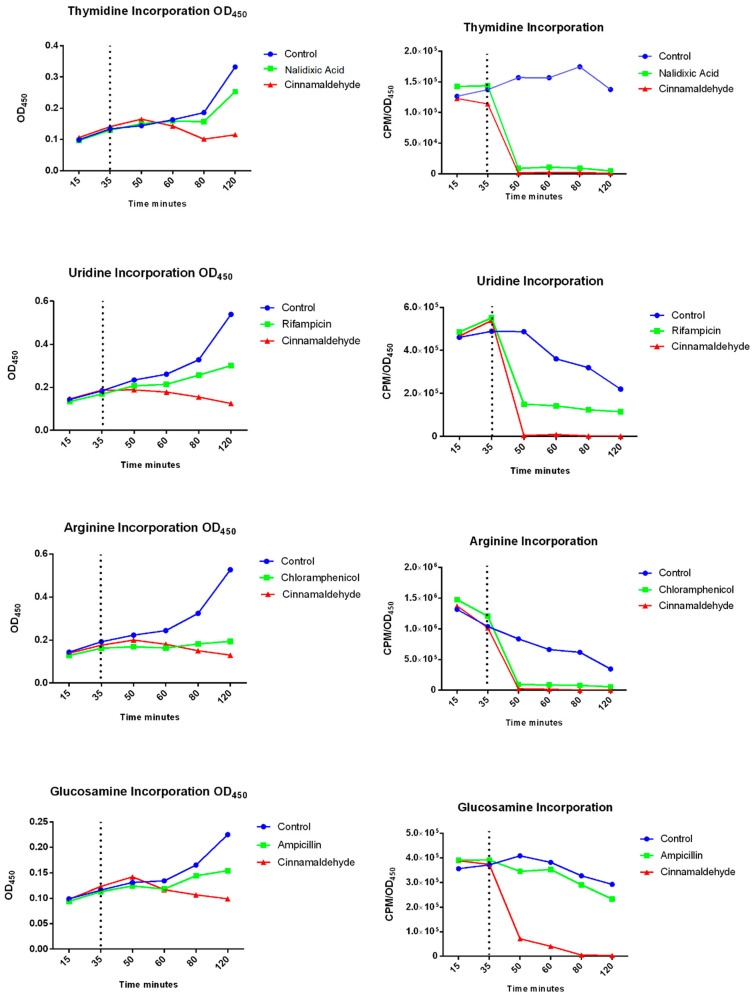
Effects of cinnamaldehyde on the assembly of the main bacterial structures by incorporating the radioactive precursors thymidine, uridine, arginine, and glucosamine that are essential for the synthesis of DNA, RNA, proteins, and cell wall, respectively. The cultures were treated with cinnamaldehyde or antibiotic after 35 min. The samples were collected at determined periods (15, 35, 50, 60, 80, 100, and 120 min) and transferred to a tube containing 0.375 μCi of each precursor. The sample was allowed to stand, according to the specific time of incorporation of each precursor. Thus, the value obtained by reading the radioactivity was subtracted from the value generated by the O.D., and determined the final result of the incorporation.

**Figure 2 biomolecules-11-00302-f002:**
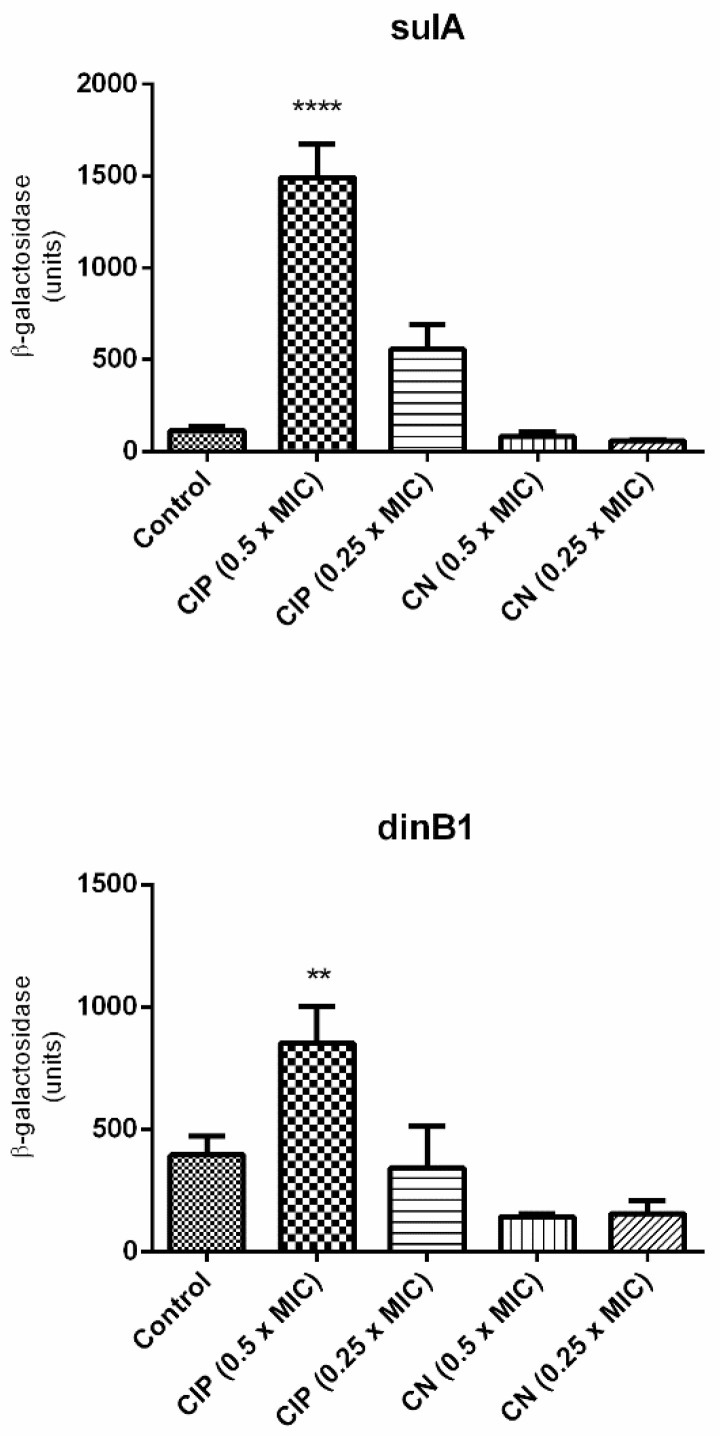
Expression of *sulA* and dinB1, genes linked to the SOS response. The test was performed using *E. coli sulA::lacZ* (ALO 4696) and *E. coli dinB1::lacZ* (ALO 562). The strains were grown in LB broth until OD_600_ of 0.1 was reached. Strains were incubated with ciprofloxacin as positive control and cinnamaldehyde. Then the supernatant was added to 1 mL of Zeta buffer containing ONPG (substrate for the enzyme β-Galactosidase). The tubes were again incubated in a 30 °C water bath with shaking and the time taken for each sample until the color change was verified. **** *p* < 0.0001 when compared to the other groups. ** *p* < 0.001 when compared to the other groups.

**Figure 3 biomolecules-11-00302-f003:**
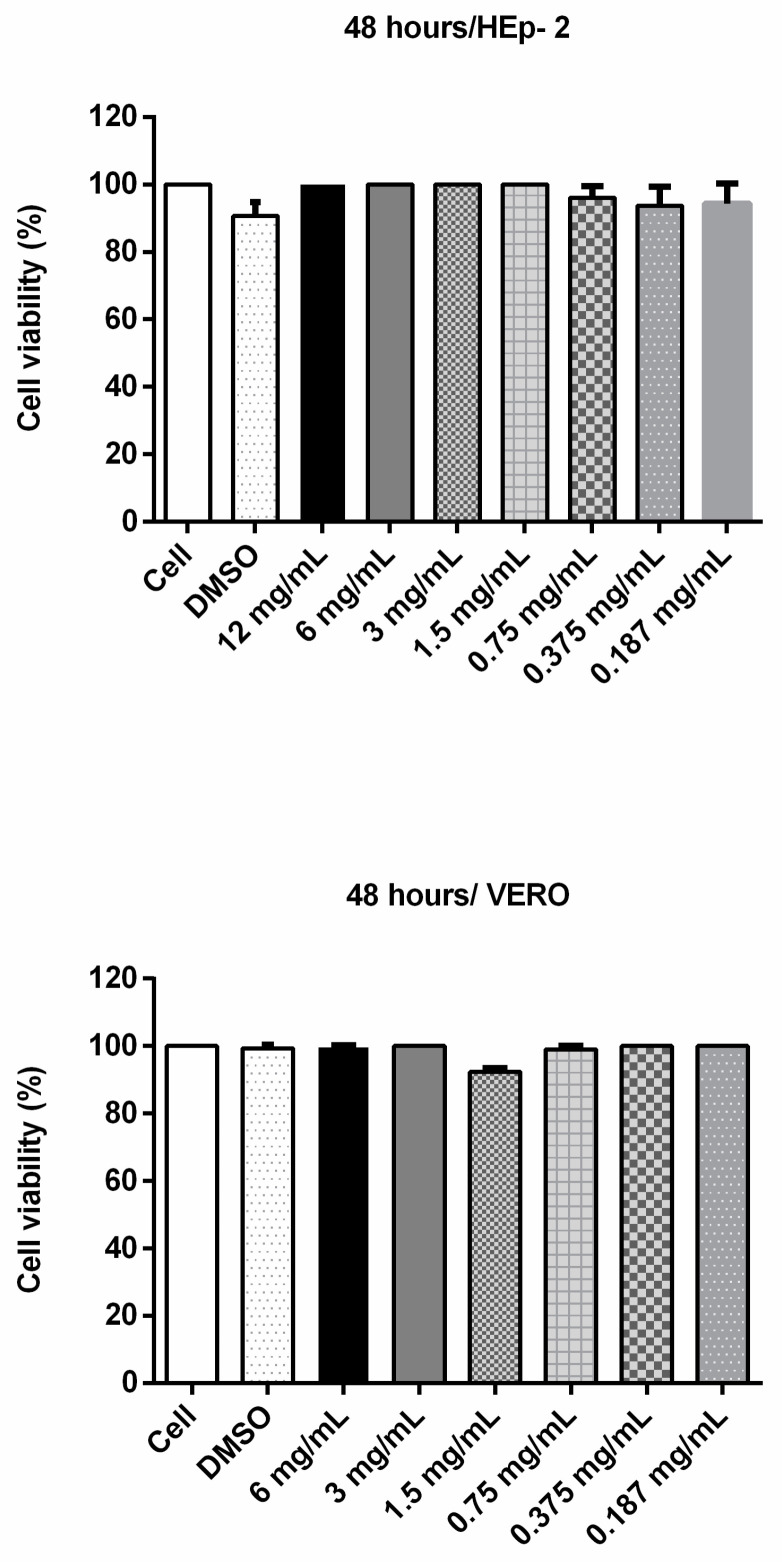
Cell viability assay (MTT) with Vero and HEp-2 cells incubated with cinnamaldehyde at different concentrations for 48 h. To evaluate a possible cytotoxic effect of cinnamaldehyde, the compound at various concentrations was incubated with VERO or Hep-2 cells for 48 h. Then 100 µL MTT (3- (4,5-dimethylthiazol-2-yl) -2,5-diphenyltetrazolium bromide) was added to each well and the plates were incubated for 3 h (37 °C and 5% CO_2_); after, 2 µL of dimethyl sulfoxide (DMSO) was added. The contents of each well were subjected to absorbance determination at a wavelength of 550 nm.

**Figure 4 biomolecules-11-00302-f004:**
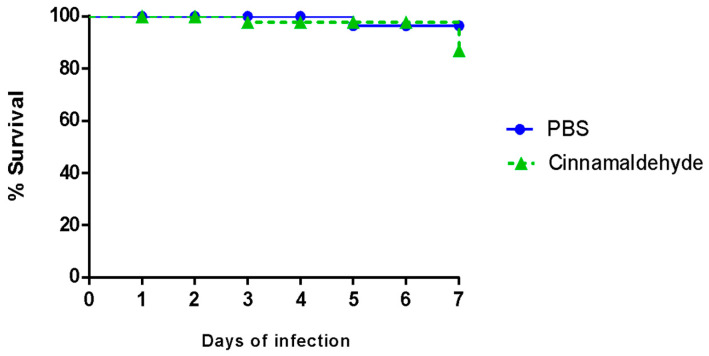
Survival curve with *Tenebrio molitor* larvae treated with PBS or cinnamaldehyde. Larvae (~100 mg) were randomly assigned to experimental groups (*n* = 10/group), and then treated with injection of 10 μL cinnamaldehyde or PBS. After injections, the larvae were incubated at 37 °C and the mortality rate was observed for seven days after infection.

**Figure 5 biomolecules-11-00302-f005:**
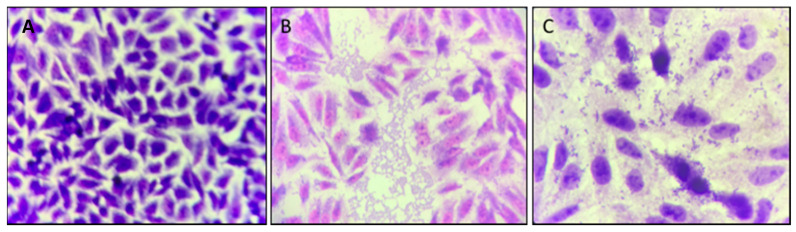
*E. coli* 042 adhesion test on HEp-2 cells exposed to cinnamaldehyde (600 µg/mL). The cells were stained with May–Grünwald / Giemsa solution and visualized by 1,000-fold light microscopy. (**A**) HEp-2 cell control, (**B**) EAEC 042 strain aggregate adhesion pattern, (**C**) strain 042 adhesion pattern after cinnamaldehyde treatment (600 µg/mL).

**Figure 6 biomolecules-11-00302-f006:**
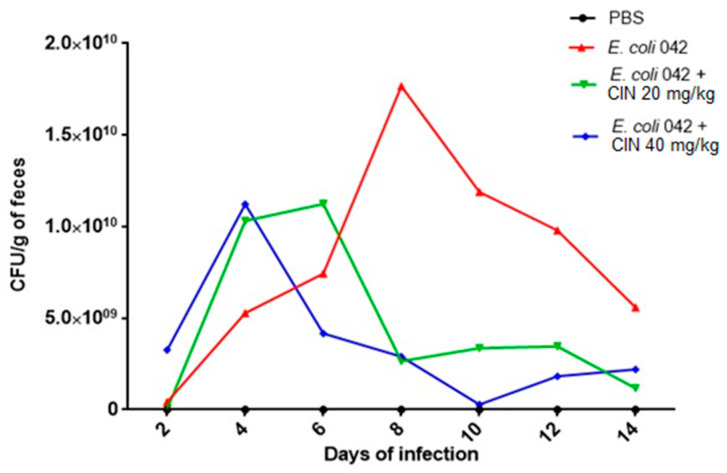
Colony forming units isolated from feces from animals colonized by EAEC 042 and treated with cinnamaldehyde. Suspensions of *E. coli* strain 042 were prepared to a final concentration of 5 × 10^3^ CFU/mL. The inoculum (200 μL) was administered orally, and fresh feces were collected daily up to 15 days after infection. Feces were weighed, homogenized in 1x sterile PBS (pH 7.4), and serial dilutions were seeded in MacConkey Agar containing streptomycin (100 μg/mL) for CFU counting.

**Table 1 biomolecules-11-00302-t001:** *Escherichia coli* strains used in the evaluation of the effect of cinnamaldehyde.

Strain	Description	Reference
*E. coli* 042	Standard strain for studies with enteroaggregative *E. coli* (EAEC), isolated from an outbreak of diarrhea in Peru.	[24]
*E. coli* HB101	Non-pathogenic strain, used as negative control of adhesion assays.	[25]
*E. coli* MG1655	Non-pathogenic wild type strain used as control in SOS and radioisotope assays.	[26] [27]
*E. coli* ALO4696	MG1655; *sulA::lac*Z.	This work
*E. coli* ALO562	GW1010 *dinB1::Mud*(Ap, lac)	[28]
*E. coli* BW25113 (ALO 4628)	Wild-type strain from Keio Collection	[29]
*E. coli* JW3682 (ALO 3542)	BW25113; *ΔyidD*::*kan*, Kan^R^	[29]
*E. coli* JW3820 (ALO 3544)	BW25113; *Δfre*::*kan*, Kan^R^	[29]
*E. coli* JW2513 (ALO 3545)	BW25113; *ΔiscU*::*kan*, Kan^R^	[29]
*E. coli* JW3879-1 (ALO 4554)	BW25113; *ΔsodA*::*kan*, Kan^R^	[29]
*E. coli* JW1648-1 (ALO 4555)	BW25113; *ΔsodB*::*kan*, Kan^R^	[29]

**Table 2 biomolecules-11-00302-t002:** Description of the precursors used in the incorporation test of radioactive macromolecules. The effects of cinnamaldehyde on the assembly of the main bacterial structures were evaluated by the incorporation of radioactive precursors thymidine, uridine, arginine, and glucosamine for DNA, RNA, protein, and cell wall synthesis, respectively.

Precursors	Marked Precursor	Function	Time of Incorporation	Antibiotic Control
Thymidine	H^3^-Thymidine	DNA Replication	4 min	Nalidixic Acid
Uridine	H^3^-Uridine	RNA Synthesis	2 min	Rifampicin
Glucosamine	H^3^-Glucosamine	Cell wall Synthesis	20 min	Ampicillin
Arginine	H^3^-Arginine	Protein Synthesis	4 min	Chloramphenicol

**Table 3 biomolecules-11-00302-t003:** Minimum Inhibitory Concentration and Minimum Bactericidal Concentration using cinnamaldehyde.

*E. coli* Strain	MIC (µg/mL)	MBC (µg/mL)
042	780	1560
HB101	780	1560
4628	1560	1560
3542	780	3120
3544	780	780
3545	780	3120
4554	3120	3120
4555	3120	3120

## Data Availability

No new data were created or analyzed in this study. Data sharing is not applicable to this article.

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
