# Peer review of "New Insights into the Antimicrobial Action of Cinnamaldehyde towards Escherichia coli and Its Effects on Intestinal Colonization of Mice"

_biomolecules, 2021, doi:10.3390/biom11020302_

Round 1
Reviewer 1 Report
The aim of this study was to evaluate the effect of cinnamaldehyde in colonization of mouse gut by pathogenic E. coli, as well as to provide more insights into its antimicrobial action mechanism.
I miss in title mice: New insights into the antimicrobial action of Cinnamaldehyde towards Escherichia coli and its effects on intestinal colonization of mice.
Line 27 replace animals as mice
Line 29 replaces new drugs as antimicrobial agents or antimicrobial substances etc.
I think that is not right: Escherichia coli is an important pathogen responsible for numerous cases of diarrhea. Please write in the concrete which E. coli. Same in all text.
In part of Mouse colonization in material and method I miss describing how animals were treated with E. coli and how animals were fed with Cinnamaldehyde. How were fresh fecal samples collected?
Line 182 describes serial dilution, please specify which dilution.
Discussion must be described because it is very general and needs more detailed comparison with results.
Conclusion is very general and needs more detailed information about Cinnamaldehyde concentration in vivo and in vitro study.
Author Response
Rebuttal letter
Manuscript: biomolecules-1095396
Title: New insights into the antimicrobial action of cinnamaldehyde towards Escherichia coli and its effects on intestinal colonization of mice
Answers to the Reviewers´ Comments
We would like to thank all comments and suggestions made by reviewer, which greatly improved the quality of the manuscript.
The aim of this study was to evaluate the effect of cinnamaldehyde in colonization of mouse gut by pathogenic E. coli, as well as to provide more insights into its antimicrobial action mechanism.
I miss in title mice: New insights into the antimicrobial action of Cinnamaldehyde towards Escherichia coli and its effects on intestinal colonization of mice.
Answer: We have modified the title as suggested.
Line 27 replace animals as mice
Answer: This correction has been made in the text.
Line 29 replaces new drugs as antimicrobial agents or antimicrobial substances etc.
I think that is not right: Escherichia coli is an important pathogen responsible for numerous cases of diarrhea. Please write in the concrete which E. coli. Same in all text.
Answer: We have modified the text as suggested.
In part of Mouse colonization in material and method I miss describing how animals were treated with E. coli and how animals were fed with Cinnamaldehyde. How were fresh fecal samples collected?
Line 182 describes serial dilution, please specify which dilution.
Answer: We have added this information in the text.
“Fresh fecal samples were collected in sterile tubes, weighted, diluted, and homogenized in sterile PBS. Serial dilutions of these preparations (1:101 until 1:106) were then plated onto MacConkey agar containing streptomycin (100 µg/mL) for determination of CFU/g.” (Lines 183-186)
Discussion must be described because it is very general and needs more detailed comparison with results.
Answer: We have modified the discussion as suggested and more details were added to the manuscript. The modifications are highlighted in yellow in the text.
Conclusion is very general and needs more detailed information about Cinnamaldehyde concentration in vivo and in vitro study.
Answer: We have modified the text as suggested.
“Taken together our data point to an important role of cinnamaldehyde as inhibitor of the growth of the pathogenic strain EAEC 042 in vitro and in vivo tests. The compound was not toxic to T. molitor larvae neither to Hep-2 and Vero cells, and reduced the adhesion of the bacterium on HEp-2 cells. In addition, it was possible to show that the compound interfered in the incorporation of key molecules for the assembly of essential structures to the bacterium. Finally, intestinal colonization of mice by EAEC 042 was reduced with cinnamaldehyde treatment (20 and 40 mg/kg). Such results show that the compound is effective in the treatment against the pathogenic strain 042 and a promising candidate for the development of novel antibacterial drugs.” (Lines 416-424)
Reviewer 2 Report
Dear Authors,
I have reviewed the manuscript entitled "New insights into the antimicrobial action of Cinnamaldehyde towards Escherichia coli and its effects on intestinal colonization" and found it to be well written and of interest for the scientific community as there is a great necessity for developing new antimicrobials.
There are some small observations that should be addressed in order to improve the quality of the manuscript. They are mentioned below:
- Please either write cinnamaldehyde with initial small letter or capital letter, but write it the same throughout the entire manuscript.
- Page 2, line 70: There is a space missing between "into" and "E. coli".
- Page 3, lines 79 and 98: Please use "reached" (past tense) instead of "reach" (present tense) as it refers to a past activity.
- Page 3, line 89: Please correct "cinnamaldehydes effect" to "cinnamaldehyde's effect".
- Page 3, line 89: I suggest the following change: "...synthesis: incorporation...", because otherwise the last sentence has no predicate.
- Page 3, line 116: Please use "grown" instead of "grow".
- Page 4, line 159: Please detail PBS the first time it appears in the manuscript.
- Several words at the end of the lines are wrongly divided into syllables. Please check and correct throughout the entire manuscript.
- Page 5, line 202: Please change "Inhibition" to "Inhibitory" as mentioned previously in the manuscript.
- Page 5, line 210: Please correct to "cinnamaldehyde's effects".
- Page 8, lines 256 and 257: Please correct "infection" to "injection".
- Page 9, line 260: I suggest the following modification: "EAEC 042 strain onto HEp-2 cells".
- Page 9, line 263: I suggest the following rephrasing: "treatment with cinnamaldehyde, it can be noted the substance's anti-adhesion effect".
- Page 9, line 276: Please use "showed" instead of "show".
- Figure 6: I recommend using the same concentrations for the cinnamaldehyde in data labels as the one mentioned in the description of the method used (20 and 40 mg/kg instead of 3 mg/mL and 6 mg/mL) to avoid any confusions.
- Page 10, line 297: Please correct to "nor was it detrimental to".
- Page 10, line 298: Please use "brought" instead of "bring".
- Page 11, line 321: Please change "affirm" to "affirmed".
- Page 11, line 331: Please change "was" (singular) to "were" (plural).
- Page 11, line 339: I suggest the following modification: "also affirmed that increasing the".
- Page 11, line 360: I suggest the following change: "essential oils, demonstrated that most of them".
- Page 11, line 362 and page 12, line 383: Please use "emphasized" instead of "emphasize".
- Page 12, line 376: I recommend the following modification: "close to zero on the last day, in this way, showing a better".
- Page 12, line 395: Cinnamaldehyde is not an essential oil, but a component of cinnamon essential oil. Please correct.
- Page 12, line 400: Please change "bacterial" to "bacteria".
- Page 12, line 407: I recommend the following modification: "in vitro and in vivo tests, was not toxic to T. molitor".
Author Response
Rebuttal letter
Manuscript: biomolecules-1095396
Title: New insights into the antimicrobial action of cinnamaldehyde towards Escherichia coli and its effects on intestinal colonization of mice
Answers to the Reviewers´ Comments
We would like to thank all comments and suggestions, which greatly improved the quality of the manuscript. All suggestions mentioned, without exception, were made in the manuscript. The modifications are highlighted in yellow in the text.
I have reviewed the manuscript entitled "New insights into the antimicrobial action of Cinnamaldehyde towards Escherichia coli and its effects on intestinal colonization" and found it to be well written and of interest for the scientific community as there is a great necessity for developing new antimicrobials.
There are some small observations that should be addressed in order to improve the quality of the manuscript. They are mentioned below:
- Please either write cinnamaldehyde with initial small letter or capital letter, but write it the same throughout the entire manuscript.
- Page 2, line 70: There is a space missing between "into" and " coli".
- Page 3, lines 79 and 98: Please use "reached" (past tense) instead of "reach" (present tense) as it refers to a past activity.
- Page 3, line 89: Please correct "cinnamaldehydes effect" to "cinnamaldehyde's effect".
- Page 3, line 89: I suggest the following change: "...synthesis: incorporation...", because otherwise the last sentence has no predicate.
- Page 3, line 116: Please use "grown" instead of "grow".
- Page 4, line 159: Please detail PBS the first time it appears in the manuscript.
- Several words at the end of the lines are wrongly divided into syllables. Please check and correct throughout the entire manuscript.
- Page 5, line 202: Please change "Inhibition" to "Inhibitory" as mentioned previously in the manuscript.
- Page 5, line 210: Please correct to "cinnamaldehyde's effects".
- Page 8, lines 256 and 257: Please correct "infection" to "injection".
- Page 9, line 260: I suggest the following modification: "EAEC 042 strain onto HEp-2 cells".
- Page 9, line 263: I suggest the following rephrasing: "treatment with cinnamaldehyde, it can be noted the substance's anti-adhesion effect".
- Page 9, line 276: Please use "showed" instead of "show".
- Figure 6: I recommend using the same concentrations for the cinnamaldehyde in data labels as the one mentioned in the description of the method used (20 and 40 mg/kg instead of 3 mg/mL and 6 mg/mL) to avoid any confusions.
- Page 10, line 297: Please correct to "nor was it detrimental to".
- Page 10, line 298: Please use "brought" instead of "bring".
- Page 11, line 321: Please change "affirm" to "affirmed".
- Page 11, line 331: Please change "was" (singular) to "were" (plural).
- Page 11, line 339: I suggest the following modification: "also affirmed that increasing the".
- Page 11, line 360: I suggest the following change: "essential oils, demonstrated that most of them".
- Page 11, line 362 and page 12, line 383: Please use "emphasized" instead of "emphasize".
- Page 12, line 376: I recommend the following modification: "close to zero on the last day, in this way, showing a better".
- Page 12, line 395: Cinnamaldehyde is not an essential oil, but a component of cinnamon essential oil. Please correct.
- Page 12, line 400: Please change "bacterial" to "bacteria".
- Page 12, line 407: I recommend the following modification: "in vitro and in vivo tests, was not toxic to molitor".
Reviewer 3 Report
Check the numbers of each and sub-section and revise according to the Guide for Authors
Cite the abbreviation of HEp2 (human epithelial type 3 cells)
Table 1. remove = from E. coli ALO562
In l. 197, you write: MIC ranging from 400 to 3,125, but in Table 3 the value 400 does not appear. Also, in Table 3 there is a value 0,750 : is this correct?
l.248:100µl - change to 100µL
l. 339: The authors affirm- write The authors affirmed
l.339: the increasing the dose- write the increasing of the dose
l.355: Here : write Here,
l.362 & l.383: The authors emphasize - The authors emphasized
Insert DOI, to the references, where possible,
Author Response
Rebuttal letter
Manuscript: biomolecules-1095396
Title: New insights into the antimicrobial action of cinnamaldehyde towards Escherichia coli and its effects on intestinal colonization of mice
Answers to the Reviewers´ Comments
We would like to thank all comments and suggestions made by reviewer, which greatly improved the quality of the manuscript.
Check the numbers of each and sub-section and revise according to the Guide for Authors
Cite the abbreviation of HEp2 (human epithelial type 3 cells)
Table 1. remove = from E. coli ALO562
Answer: All the modifications were made in the manuscript.
In l. 197, you write: MIC ranging from 400 to 3,125, but in Table 3 the value 400 does not appear. Also, in Table 3 there is a value 0,750: is this correct?
Answer: You are correct. In fact, the value 400 has been rounded in the first version of the manuscript, but has already been adjusted to the original value 390. In relation to the value 750, the correct is 780. The table 3 has been corrected. Thank you very much for your observation.
l.248:100µl - change to 100 µL
- 339: The authors affirm - write The authors affirmed
l.339: the increasing the dose- write the increasing of the dose
l.355: Here: write Here,
l.362 & l.383: The authors emphasize - The authors emphasized
Answer: All the modifications were made in the manuscript. The changes are highlighted in yellow in the text.
Insert DOI, to the references, where possible.
Answer: The missing DOI were inserted to the references.
Round 2
Reviewer 1 Report
All comments were accepted.